TOPICAL REVIEW

# Exosomes and the cardiovascular system: role in cardiovascular health and disease

Karla B. Neves[1] (iD), Francisco J. Rios[1], Javier Sevilla-Montero[2], Augusto C. Montezano[1] and Rhian M. Touyz[1,3] (iD)

[1]*Institute of Cardiovascular and Medical Sciences, University of Glasgow, UK*
[2]*Biomedical Research Institute La Princesa Hospital (IIS-IP), Department of Medicine, School of Medicine, Universidad Autónoma of Madrid (UAM), Madrid, Spain*
[3]*Research Institute of the McGill University Health Centre (RI-MUHC), McGill University, Montreal, Canada*

Handling Editors: Ian Forsythe & Harold Schultz

The peer review history is available in the Supporting information section of this article (https://doi.org/10.1113/JP282054#support-information-section).

The Journal of Physiology

The Journal of
**Physiology**

**Abstract** Exosomes, which are membrane-bound extracellular vesicles (EVs), are generated in the endosomal compartment of almost all eukaryotic cells. They are formed upon the fusion of multivesicular bodies and the plasma membrane and carry proteins, nucleic acids, lipids and other cellular constituents from their parent cells. Multiple factors influence their production including cell stress and injury, humoral factors, circulating toxins, and oxidative stress. They play an important role in intercellular communication, through their ability to transfer their cargo (proteins, lipids, RNAs) from one cell to another. Exosomes have been implicated in the pathophysiology of various diseases including cardiovascular disease (CVD), cancer, kidney disease, and inflammatory conditions. In addition, circulating exosomes may act as biomarkers for diagnostic and prognostic strategies for several pathological processes. In particular exosome-containing miRNAs have been suggested as biomarkers for the diagnosis and prognosis of myocardial injury, stroke and endothelial dysfunction. They may also have therapeutic potential, acting as vectors to deliver therapies in a targeted manner, such as the delivery of protective miRNAs. Transfection techniques are in development to load exosomes with desired cargo, such as proteins or miRNAs, to achieve up-regulation in the host cell or tissue. These advances in the field have the potential to assist in the detection and monitoring progress of a disease in patients during its early clinical stages, as well as targeted drug delivery.

(Received 30 November 2021; accepted after revision 15 March 2022; first published online 20 March 2022)

**Corresponding authors** Karla B. Neves and Francisco J. Rios: Institute of Cardiovascular and Medical Sciences, BHF Glasgow Cardiovascular Research Centre, University of Glasgow, 126 University Place, Glasgow G12 8TA, UK. Email: karla.neves@glasgow.ac.uk and francisco.rios@glasgow.ac.uk

**Abstract figure legend** Schematic representation of an exosome and its cargo carrying a diverse number of bioactive factors and signalling molecules that have been implicated in many cardiovascular diseases, including pulmonary hypertension, brain injury and stroke, vascular diseases, and renal and heart diseases. We have highlighted some of the physiological and pathological roles of the exosomes, which have been implicated not only in beneficial effects such as renal and cardiac protection, but also in the pathophysiology of various cardiovascular diseases, including hypertension, kidney failure and ischaemic heart disease. The role of exosomes as non-invasive biomarkers for diagnostic and prognostic strategies throughout disease development have also been addressed, particularly the exosome-containing miRNAs which have been suggested as biological markers of myocardial injury, stroke, acute kidney injury and endothelial dysfunction. Finally, the therapeutic potential of exosomes is also discussed, especially mechanisms mediated by transferring protective miRNAs and transfection techniques in development to load exosomes with a desired cargo (e.g. proteins, miRNAs) to achieve up-regulation in the host cell or tissue.

## Introduction

Exosomes belong to the family of extracellular vesicles (EVs) and range in size from 30 to 150 nm. They are released by different cell types, including those of the cardiovascular system, and are found in various body fluids such as blood, urine, saliva, amniotic fluid and breast milk. Exosomes are rich in bioactive molecules, including DNA, microRNAs (miRNAs), messenger RNAs (mRNAs) and proteins. Their ability to transfer their bioactive cargo between cells suggests that they are crucial players in intercellular communication (Weber et al., 2021; Zara et al., 2020). Current evidence indicates that exosomes can mediate autocrine, paracrine and endocrine functions. Paracrine or endocrine intercellular trafficking of proteins/genetic material seems to play an important role in intercellular communication (Record et al., 2014;

**Karla B. Neves** is a research associate in the Institute of Cardiovascular and Medical Sciences at the University of Glasgow (UK), working on the vascular and molecular biology of hypertension. She completed her MSc and PhD in Sciences at Sao Paulo University (Brazil) and was awarded a bachelor's degree in nursing at the Federal University of Mato Grosso (Brazil). **Francisco J. Rios** is a research fellow in the Institute of Cardiovascular and Medical Sciences at the University of Glasgow (UK), working on the importance of the inflammatory mechanisms related to organ damage associated with cardiovascular diseases.

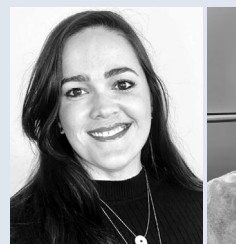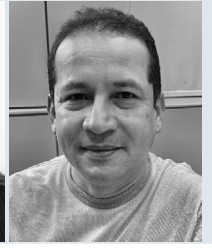

Wood et al., 2000; Xu et al., 2013; Yuan et al., 2018; Zhang, Lu et al., 2019).

There is increasing interest in the potential role of exosomes in the clinic. The number, origin and cargo of circulating exosomes differ under pathological conditions, suggesting their potential as a biomarker of disease. Many studies have shown associations between circulating exosomes and risk and severity of disease including CVD, diabetes, kidney disease and cancer (Barros & Carvajal, 2017; Eirin & Lerman, 2021; Medeiros et al., 2020; Mitchell et al., 2009; Osaki & Okada, 2019; Wang et al., 2021; Weber et al., 2021; Yang et al., 2021; Zara et al., 2020; Zhang et al., 2020; Zhang, Liang et al., 2021). However, there is a paucity of studies relating to exosome phenotyping in CVD patients with distinct underlying disease severity, and the pathophysiological role of exosomes in CVD still remains unclear (Ma et al., 2021; Zamani et al., 2019; Zara et al., 2020). In this review we summarize recent findings about the function of exosomes and their role in crosstalk between different cell types and their potential utility as diagnostic/prognostic biomarkers and therapeutic agents in CVD.

**Exosome biogenesis and biology.** Subtypes of exosomes are based on various morphometric characteristics, including size. Exosomes (termed EVs) range in size from 30 to 150 nm and are released from cells in a complex exocytic process that is dependent on several intracellular activation pathways. They are different from microvesicles (MVs), also known as microparticles, which range in size from 100 nm–1 μm, and apoptotic bodies, which are larger structures of 0.5–5 μm diameter (van Niel et al., 2018). Exosomes are potentially produced by all cells in the body and can be found in body fluids, including plasma, urine, breast milk, cerebrospinal fluid and, experimentally, in cell culture medium (van Niel et al., 2018). They carry proteins, lipids, phospholipids, hormones and acid nucleic, including mRNA, miRNA, long non-coding RNA (lncRNA) (de Abreu et al., 2020; van Niel et al., 2018). Exosomes are formed by the inward budding of the plasma membrane whereas MVs are formed by the outward blebbing of the plasma membrane (Fig. 1). Although exosomes and MVs can carry similar cargo, it is still unclear which component of their cargo best distinguishes these different types of EV (Lee et al., 2018).

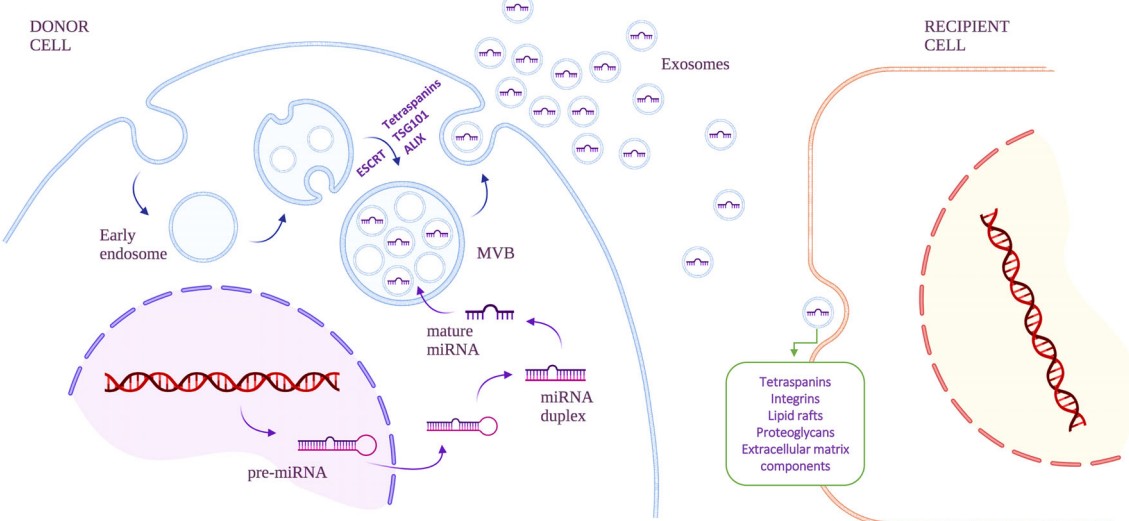

**Figure 1. Schematic illustration of the exosome biogenesis and exosomal micro-RNA (miRNA) synthesis and loading**
The formation of exosomes is related to the endosomal trafficking pathway of the donor cell, beginning with the invagination of the plasma membrane to form an early endosome. Exosomes biogenesis then requires additional inward budding processes affecting the early endosome membrane, giving rise to the multivesicular body (MVB). For miRNA to be loaded into the MVB, pre-miRNA transcripts are exported from the nucleus to the cytoplasm, where they undergo a series of additional modifications. First, pre-miRNAs are cleaved into double-strand miRNAs and then single-strand mature miRNA are loaded into the internal vesicles of the MVB. The release towards the extracellular medium of mature exosomes containing their specific cargo, including miRNA, is accomplished through the fusion of the MVB external membrane and the plasma membrane of the donor cell. Once in the extracellular milieu, exosomes may be captured, through new endocytosis events, by surrounding recipient cells in a paracrine way, but may also be released into the bloodstream and exert long-distance effects away from their original cellular sources.

The exact molecular mechanisms involved in the biogenesis of exosomes are not completely elucidated but involve many phases: (1) early endosome – this starts with the invagination of the plasma membrane forming endocytic vesicles. They are located in the periphery of the membrane and are characterized by the presence of RAB5A, a GTPase that binds to a variety of proteins and regulates intracellular membrane trafficking (Jopling et al., 2009); (2) maturation to the late endosome – this is followed by reducing levels of RAB5A and increased expression of RAB7A (Jopling et al., 2009), a reduction in pH and cytosolic $Ca^{2+}$ levels (Bose et al., 2021), triggering an increase in multivesicular bodies (MVBs), which are formed by intraluminal vesicles (ILVs) (de Abreu et al., 2020; van Niel et al., 2018); (3) late endosomes – these can either fuse with lysosomes to be degraded or are released to the extracellular environment as exosomes (Fig. 1).

The balance of RAB5A and RAB7A is controlled by NOTCH4 (Jin et al., 2021). In early endosomes, NOTCH4 activation increases expression of the Ras-related protein RAB5A, whereas in the maturation process, NOTCH4 signalling is reduced, causing a reduction of RAB5A and increased RAB7A (Jin et al., 2021). Components of the endosomal sorting complex required for transport (ESCRT) are important regulators during the formation of MVBs and ILVs. ESCRT is usually involved in cell membrane repair and is highly relevant in exosome biology. ESCRT depletion results in inhibition of exosome biogenesis (Colombo et al., 2013). The participation of ESCRT in exosome biogenesis occurs in phases. Initially, ESCRT-0 targets early endosomes. This is followed by activation of ESCRT-I and -II, which are involved in membrane deformation and budding, which sequesters the parent cell cargo. ESCRT-III drives vesicle scission (Colombo et al., 2013). The complex syndecan-4/syntenin/ALIX also plays a role in exosome formation (Addi et al., 2020; Baietti et al., 2012). Mechanisms independent of ESCRT have also been described and tetraspanin protein families, such as CD63, CD82 and CD9, may play an important function. Tetraspanins are present on the surface of exosomes, and can therefore also be used as their biomarkers (van Niel et al., 2011). Other biomarkers of exosomes include TSG101, syntenin-1 and ALIX (de Abreu et al., 2020; van Niel et al., 2018).

Different types of exosomes carrying different cargoes can be derived from the same cell, depending on the location of the cell membrane from which the exosome is formed (Willms et al., 2016). *In vitro*, differential surface markers are evident. For example, in cultured cardiomyocytes 30% of the exosomes exhibit caveolin-3 and 80% flotillin-1 (Waldenstrom et al., 2012). Additionally, besides the plasma membrane invagination, several of the ILVs carry endogenous molecules originating from the endoplasmic reticulum-Golgi network, which contribute to heterogeneity of the cargo (Saadi et al., 2018). Proteins within the exosome cargo can also exhibit post-translationally modified motifs, such as glycosylation and ubiquitination (Ageta et al., 2018; Szabo-Taylor et al., 2015).

In the extracellular space, exosomes interact with target cells and deliver their content, which in turn triggers changes in target cell signalling and function. Mechanisms underlying the interaction between exosomes and target cells include fusion, endocytosis (clathrin-mediated or caveolin-mediated) or phagocytosis (de Abreu et al., 2020; van Niel et al., 2018). A variety of membrane components participate in the interaction between exosomes and target cells, including tetraspanins, integrins, lipids, lectins, heparan sulphate, proteoglycans and extracellular matrix components. The cell population and phenotype of the target cell also dictates the nature of the interaction with the exosomes (Horibe et al., 2018). In the inflammatory response, for example, cells exhibit higher receptor expression and increase the endocytosis index, hence these cells can interact with a higher number of exosomes and MVs (Svensson et al., 2013).

Based on the exosome database, ExoCarta and the International Society for Extracellular Vesicles, more than 41,860 proteins, 7540 RNAs and 1116 lipid molecules have been identified and validated in exosomes (Keerthikumar et al., 2016). Accordingly, the activation phenotype of the parent cell is crucial in the characteristics of the cargo of exosomes and influences how target cells respond. Interaction does not occur only in cells from the same lineage; exosomes that originate from one cell population can change the activation phenotype of different cell types (Waldenstrom et al., 2012).

**Exosomes as mediators of cardiovascular health and disease.** The cardiovascular system involves a complex network of different cell types, including cardiomyocytes, cardiac progenitor cells, endothelial cells (ECs), vascular smooth muscle cells (VSMCs), fibroblasts, pericytes, adipocytes and immune cells. The crosstalk and intercellular communication between these cells are not only responsible for the maintenance of the homeostasis and function of this system but also account for some pathological processes in CVD. Exosomes may play a role in cellular crosstalk, especially via their paracrine and endocrine functions (Record et al., 2014; Wood et al., 2000; Xu et al., 2013; Yuan et al., 2018; Zhang, Lu et al., 2019) (Fig. 2).

*Cardiovascular health and protection.* Some studies have suggested that exosomes have cardioprotective effects (Liao et al., 2021; Ren et al., 2020; Sindi et al., 2020; Weber et al., 2021; Zamani et al., 2019; Table 1). Recent findings suggested that exosomes derived from

induced pluripotent stem cells (iPSCs) can transfer cardioprotective miRNAs to cardiomyocytes. miR-21 and miR-210 levels are significantly increased in iPSC-exosomes, and the transfer of iPSC-exosomes to H9C2 cells protects the cells from $H_2O_2$-induced oxidative stress via inhibition of caspase 3/7 activation. These anti-apoptotic effects of iPSC-exosomes were confirmed in a mouse model of acute myocardial ischaemia/reperfusion (Wang et al., 2015). In an acute myocardial infarction rat model, intramyocardial injections of mesenchymal stem cell (MSC)-EVs have also reulted in a significant recovery in blood flow, which was followed by a reduction in the infarct size and cardiac systolic and diastolic function preservation (Bian et al., 2014). MSC-exosomes were additionally able to protect myocardial cells from apoptosis and promote angiogenesis, leading to an improvement in cardiac systolic function in an acute myocardial infarction model (Zhao et al., 2015). Exosomes derived from endothelial progenitor cells (EPCs) have been shown to accelerate re-endothelialisation and wound healing in diabetic rats by promoting angiogenesis (Zhang et al., 2016), while exosomes derived from adipose-derived stem cells reduced injury to endothelial cells by increasing

eNOS and reducing reactive oxygen species (ROS) and inflammasome activation induced by high glucose in an experimental model of type 2 diabetes (Zhang, Jiang et al., 2021). Likewise, a protective role of exosomes for prevention or reduction of acute kidney injury has also been demonstrated (Weber et al., 2021). MSC-released exosomes improve renal function in a gentamicin and cisplatin-induced acute kidney injury models, mediated by RNA carried by the exosomes and microvesicles (Reis et al., 2012; Zhou et al., 2013).

*Cardiovascular diseases.* Exosomes derived from cardiac, endothelial and VSMCs harbour a range of miRNAs, which may be transferred to recipient cells, leading to changes in their function (Table 2). Exosomal miRNAs have been shown to play a deleterious role in CVD, including atherosclerosis, preeclampsia, hypertension, heart failure and ischaemic heart disease. In patients with peripheral artery disease (PAD), exosomes carry proinflammatory factors that directly modulate the phenotype of VSMCs and ECs (Sorrentino et al., 2020). VSMC-derived exosomes transfer miR-155 from smooth muscle to endothelial cells, damaging the tight junction of ECs, leading to increased endothelial cell

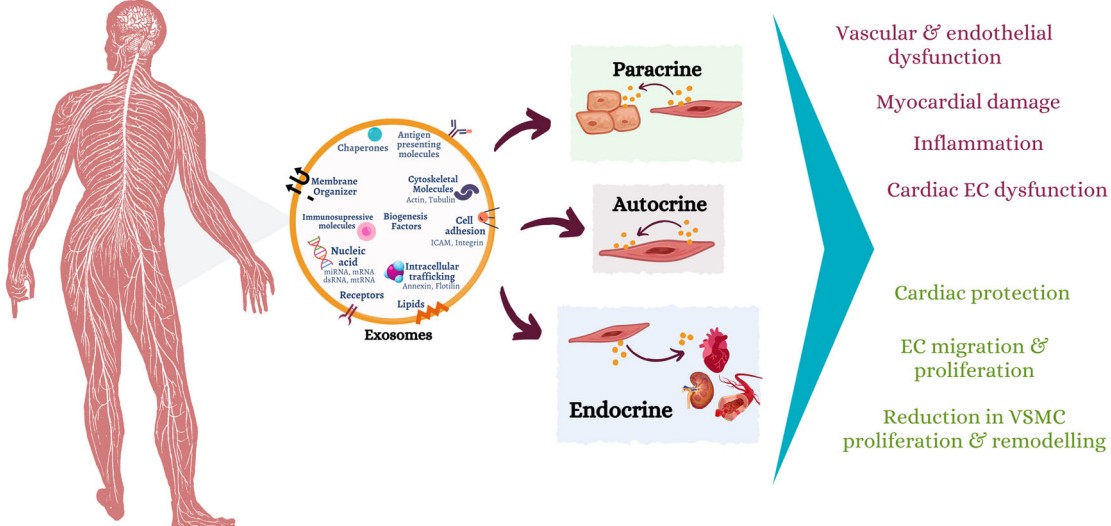

**The Journal of Physiology**

**Figure 2. Exosome-mediated intercellular communication**
Exosomes carry several bioactive factors and signalling molecules that have been implicated in many cardiovascular diseases, including pulmonary hypertension, brain injury and stroke, vascular diseases, and renal and heart diseases. They can mediate autocrine, paracrine and endocrine functions. Paracrine or endocrine intercellular trafficking of proteins and miRNAs seems to play an important role in intercellular communication. And therefore, exosomes have been implicated not only in beneficial effects such as cardiac protection, endothelial cell migration and proliferation, and reduction in VSMC proliferation and remodelling, but also in inflammatory responses, myocardium damage and vascular and endothelial cell dysfunction.

**Table 1. miRNA with beneficial/protective effects in CVD**

| Exosomalmicro-RNA | Related CVD | Biological effects | References |
|---|---|---|---|
| miR-19a | Myocardial infarction | Decreases cardiomyocytes apoptosis | Weber et al., 2021 |
| miR-21 | Myocardial infarction | Decreases cardiomyocytes apoptosis | Wang et al., 2015 |
| miR-21-5p | Myocardial infarction | Decreases cardiac EC apoptosis | Liao et al., 2021 |
| miR-22 | Myocardial infarction | Increases cardiomyocytes survival, decreases cardiac fibrosis | Weber et al., 2021 |
| miR-126 | Diabetic ischaemic stroke | Decreases EC apoptosis, increases EC migration and tube formation capacity | Wang et al., 2020 |
| miR-126 | Myocardial infarction | Increases EC migration, increases VEGFR2 pathway, decreases cardiac fibrosis | Luo et al., 2017 |
| miR-133 | Cardiac fibrosis | Promotes cardiac fibroblasts mesenchymal-endothelial transition | Lin et al., 2019 |
| miR-144-5p | Intracranial aneurysm | Decreases EC apoptosis and vascular remodeling | Yang et al., 2021 |
| miR-155-5p | Hypertension | Decreases VSMC proliferation and vascular remodeling | Ren et al., 2020 |
| miR-181-5p,miR-324-5p | Pulmonary hypertension | Decreases PAEC apoptosis and vascular muscularization | Sindi et al., 2020 |
| miR-210 | Myocardial infarction | Increases EC proliferation, migration, and tube formation capacity, decreases cardiomyocyte apoptosis and cardiac fibrosis, promotes myocardial contractility | Wang et al., 2015 |
| miR-378 | Cardiac fibrosis | Decreases cardiac fibroblasts proliferation and collagen production | Yuan et al., 2018 |

**Table 2. miRNA with deleterious effects in CVD**

| Exosomal micro-RNA | Related CVD | Biological effects | References |
|---|---|---|---|
| miR-19a-3p | Myocardial infarction | Decreases EC proliferation, increases EC apoptosis | Gou et al., 2020 |
| miR-135a-5p | Hypertension | Increases VSMC proliferation | Tong et al., 2021 |
| miR-143 | Myocardial infarction | Decreases EC proliferation and EC tube formation capacity | Geng et al., 2020 |
| miR-143 | Pulmonary hypertension | Increases EC proliferation and EC tube formation capacity, increases PASMC migration, decreases PASMC apoptosis | Deng et al., 2015 |
| miR-155 | Atherosclerosis | Decreases EC proliferation and migration, disrupts EC tight junctions | Zheng et al., 2017 |
| miR-185 | Coronary artery disease | Decreases reendothelialization | Si et al., 2021 |
| miR-200a-3p | Cardiac EC dysfunction | Increases EC apoptosis, blocks eNOS-NO, and VEGF-A pathways | Ranjan et al., 2021 |
| miR-939-5p | Myocardial infarction | Decreases EC proliferation and EC tube formation capacity, blocks iNOS-NO pathway | Li et al., 2018 |

permeability and accelerating atherosclerosis (Zheng et al., 2017). Exosomes carry functionally active endothelial nitric oxide synthase (eNOS). In preeclampsia, exosome eNOS is downregulated leading to decreased nitric oxide (NO) bioavailability (Motta-Mejia et al., 2017). Similarly, in preeclampsia, EC-derived exosomes have been demonstrated to disrupt EC function by targeting 3-hydroxy-3-methylglutaryl-CoA synthase 1 (HMGCS1), which may cause vascular disorders in the offspring (Ying et al., 2021).

In coronary artery disease (CAD), exosomes have been associated with endothelial injury and inflammation, via increasing production of inflammatory factors such as IL-1$\beta$, TNF-$\alpha$ and ICAM-1, which may, in turn,

cause endothelial dysfunction and potentially worsen the development and progression of CAD (Zhang, Liang et al., 2021). Exosomal miR-185 transfer from VSMCs to ECs was shown to impair re-endothelialisation following carotid artery injury (Si et al., 2021). Additionally, cardiac fibroblast-derived exosomes enriched with miR-200a-3p have been associated with endothelial dysfunction, via changes in the VEGF-A/PIGF signalling cascade, suggesting new mechanisms underlying endothelial dysfunction during cardiac fibrosis (Ranjan et al., 2021). Exosomes released by microvascular ECs also led to NF-$\kappa$B activation in ECs, promoting activation of inflammatory signalling and endothelial dysfunction (Migneault et al., 2020). Similarly, lung fibroblasts isolated from patients with pulmonary hypertension induce inflammatory responses and metabolic changes in macrophages (Kumar et al., 2021). VSMC-derived exosomes from spontaneously hypertensive rats (SHRs) induce proliferation in VSMCs from normotensive Wystar Kyoto (WKY) rats by mechanisms dependent on miR135a-5p (Tong et al., 2021). Moreover, exosomes from adventitial fibroblasts from SHR rats were shown to transfer angiotensin-converting enzyme (ACE) to VSMCs, which increased the levels of angiotensin II and induced cell migration by activation of AT1 receptors (Tong et al., 2018) (Table 2).

Exosomes may also contribute to atherosclerosis development and progression. Endothelial cells-derived exosomes, upon activation of CD137 inflammatory signalling, induce a proliferative and migratory phenotype in VSMC, and intimal hyperplasia after arterial injury, leading to plaque formation (Li et al., 2020). Similarly, macrophage foam cell-derived exosomes transfer integrins to VSMCs, thus promoting their migration and activation of downstream signalling pathways (Niu et al., 2016).

**Exosomes as biomarkers of cardiovascular injury.** Although the understanding of the pathophysiology of CVD has advanced in the last decades, there is still a need for novel biomarkers to predict and monitor disease risk and progression. Recent evidence suggests that exosomes may be important biomarkers in CVD. The origin, number and cargo of exosomes vary under pathological conditions, which further strengthens their potential as a specific biomarker of disease. An important advantage of exosomes as disease biomarkers is that they can travel in several body fluids to reach other cells/tissues/organs and are implicated in several pathophysiological processes throughout disease development (Table 2).

In various clinical settings, elements carried by exosomes, show potential as prognostic and diagnostic markers of CVD, where the cargo of exosomes is altered according to the severity of the disease (Medeiros et al., 2020; Weber et al., 2021; Zara et al., 2020). Exosomes and circulating miRNAs were shown to be significantly increased after myocardial ischaemia-reperfusion injury in pigs, and these exosomes accumulated miRNA-133b, miRNA-208b and miRNA-499 (Deddens et al., 2016). Exosomes containing miRNAs have been suggested as biomarkers for the diagnosis and prognosis of CVD. miRNAs of heart-derived exosomes can be detected in urine, representing a significant advance in liquid biopsy in cardiovascular medicine (Ma et al., 2021; Weber et al., 2021; Zara et al., 2020). In doxorubicin treated-cardiomyocytes, exosomes containing isoforms of glycogen phosphate were detectable earlier than troponin, which is a classic marker of cardiac injury, and therefore might be an interesting indicator of the early stages of cardiac injury (Yarana et al., 2018). This is especially important because anti-cancer drugs have been widely associated with cardiotoxicity and CVD (Neves et al., 2019; van Dorst et al., 2021). This is a field where exosomes may have clinical utility as biomarkers of disease.

Patients with myocardial infarction exhibit an increase in exosomes (Cheow et al., 2016). A recent study demonstrated that 50 exosomal miRNAs are dysregulated in patients with myocardial infarction. Among them, miR-183 expression was shown to be significantly upregulated in these patients compared with counterpart control individuals, and its expression was positively associated with the degree of myocardial injury (Zhao et al., 2019). In line with this, exosomes seem to play a role in cardiac fibrosis and remodelling following myocardial infarction, and accordingly may be biomarkers in early diagnosis of hypertrophic heart diseases (Kuwabara et al., 2011; Zara et al., 2020). They have also been implicated in the pathophysiology of diabetic cardiomyopathy (Salem & Fan, 2017) and renal fibrosis (Kuwabara et al., 2011) (Table 3).

Exosome-mediated neuroinflammation has also been suggested as an important mechanism for brain injury development, and the cargo of exosomes may represent promising biomarkers of poor outcome and neurological deficit after these injuries (Weber et al., 2021). Additionally, exosome content has been suggested as a biomarker in acute ischaemic stroke, where patients show an increase in serum concentration of exosomes and exosomal levels of the brain-specific miRNAs miR-9 and miR-124. This rise in miR-9 and miR-124 was positively correlated to infarct volumes, which suggests the potential use of exosomes to assess the degree of damage caused by ischaemic injury (Ji et al., 2016; Zara et al., 2020).

Exosomes and their content are also a suitable source of urinary biomarkers in kidney disease (Barros & Carvajal, 2017; Medeiros et al., 2020; Weber et al., 2021). Proteomic studies of urinary exosomes revealed that they contain numerous renal proteins and transporters

**Table 3. miRNA serving as CVD biomarkers**

| Exosomal micro-RNA | Related CVD | References |
| --- | --- | --- |
| miR-9, miR-124, miR-125b-2-3p, miR-134, miR-223, miR-422a | Ischaemic stroke | Chen et al., 2017; Ji et al., 2016; Li et al., 2017; Zhou et al., 2018 |
| miR-30e, miR-92a | Atherosclerosis | Ma et al., 2021; Wang et al., 2019 |
| miR-1, miR-133b, miR-183, miR-208, miR-208b, miR-499 | Myocardial infarction | Deddens et al., 2016; Ma et al., 2021; Zhao et al., 2019 |
| miR-21, miR-92a, miR-126, miR-143, miR-181b, miR-221 | Peripheral artery disease | Sorrentino et al., 2020 |
| miR-517-5p, miR-520a-5p, miR-525-5p | Gestational hypertension and preeclampsia | Hromadnikova et al., 2019 |

(e.g. thiazide-sensitive sodium chloride cotransporter, NCC; Na-K-Cl cotransporter, NKCC2; epithelial sodium channel, ENaC) (Gonzales et al., 2009). Urinary exosomes may also be useful in the diagnosis of mineralocorticoid-induced hypertension (Barros & Carvajal, 2017) and primary aldosteronism (Castagna et al., 2015). Additionally, a positive correlation between the renal-tissue expression of NCC and NKCC2 and the expression of these same proteins in urinary exosomes has been demonstrated in rat models of sodium imbalance (Esteva-Font et al., 2010). miRNAs in urinary exosomes were correlated with early and late (fibrotic) stages of acute kidney injury in an ischaemia/reperfusion injury rat model (Sonoda et al., 2019). In patients with sepsis-induced acute kidney injury there are also changes in exosomes cargo, with greater expression of neutrophil gelatinase-associated lipocalin (NGAL) (Panich et al., 2017). In end-stage chronic kidney disease, over-expression of exosome-derived miR-133, which is linked to inflammation and renal endothelial dysfunction, has been reported (Cavallari et al., 2019). Not only exosomes, but EVs in general have been suggested as direct early biomarkers of renal injury (Medeiros et al., 2020). Interestingly, microparticles have been also suggested as biomarkers and mediators of vascular dysfunction and hypertension commonly observed in cancer patients under treatment with vascular endothelial growth factor (VEGF) inhibitors (Neves et al., 2019).

**Exosomes as therapeutic agents in cardiovascular diseases.** Structural features of exosomes, such as low immunogenicity and toxicity and high ability to cross the cell membrane and tissue barriers, are attractive to research and development programmes focusing on drug delivery systems. This can be achieved by using exosomes in their native form or in combination with an additional therapeutic approaches. Considering that the activation phenotype of the parent cell influences the exosome cargo and subsequently the effects in the target cell, exosomes from various sources have been employed to provide protection against inflammatory responses and vascular damage. Native exosomes derived from EPCs improve tissue repair by reducing the production of ROS, expression of adhesion molecules and proinflammatory cytokines, including TNF-$\alpha$ and IL-6, and increasing eNOS expression (Wu et al., 2016; Zhang, Lu et al., 2019; Zhou et al., 2019). EPC-derived exosomes were also shown to reduce fibrosis by reducing endothelial-to-mesenchymal transition (Ke et al., 2017). On the other hand, in inflammatory conditions EPC-derived exosomes can induce cellular damage by increasing levels of integrin-linked kinase (ILK), which activates the NF-$\kappa$B pathway in the target cell, leading to tissue damage in a model of cardiac injury (Yue et al., 2020). Moreover, exosomes from normotensive WKY rats reduced vascular angiotensin-converting enzyme (ACE) expression and attenuated vascular remodelling and hypertension in SHR rats via mechanisms dependent on miR155-5p (Ren et al., 2020).

Promising data regarding the therapeutic use of native exosomes come from MSCs, which have been shown to be protective in cardiovascular diseases, including pulmonary arterial hypertension (Zhang et al., 2020), aneurysm (Yang et al., 2021), chronic heart failure (Wang et al., 2021) and kidney disease (Eirin & Lerman, 2021). Mechanisms are mainly mediated by transferring protective miRNAs, including miR-144-5p (Yang et al., 2021), miR-1246 (Wang et al., 2021), miR-181-5p and miR-324-5p (Sindi et al., 2020). However, identification of the intracellular targets of these protective miRNAs remains unclear. Recently, clinical trials have addressed the use of exosomes derived from allogeneic bone marrow mesenchymal stem cells to treat severely infected COVID-19 patients. A single dose of the treatment proved safe and resulted in a significant improvement in clinical manifestations and recovery in 71% of the patients (Sengupta et al., 2020). However, to date, no clinical trials have investigated the therapeutic role of exosomes in CVD.

Another potential strategy is to use exosomes in combination with other therapeutic approaches, such as exosomes loaded with small molecules or gene-editing tools (O'Brien et al., 2020). However, the main challenge is how to effectively load these exosomes without affecting their main structure and therefore damaging their ability to interact with target cells. Electroporation is an example of a commonly applied exosome loading technique that has been used to add anti-inflammatory compounds to exosomes from macrophages; the loading efficacy was around 20% and was sufficient to reduce inflammation in acute peritonitis and atherosclerosis development (Wu et al., 2020). Transfection techniques are also used to load exosomes with a desired cargo, such as proteins or miRNAs, to achieve up-regulation in the host cell or tissue. Recent findings showed that exosomes loaded with syndecan-1 ameliorated pulmonary oedema and reduced pro-inflammatory cytokines IL-1$\beta$, TNF-$\alpha$ and IL-6 (Zhang, Guo et al., 2019). Additionally, exosomes derived from cardiac stromal cells from patients with heart failure exhibited reduced regenerative effects in myocardial infarction models, which were reversed when the same exosomes were loaded with miR-21-5p, a miRNA downregulated in heart failure (Qiao et al., 2019).

The field of exosome research is growing and with advanced techniques in phenotyping and isolating different types of exosomes in the context of clinically phenotyped patients there is great promise for the possible use of exosomes to treat diseases with lower side effects and immunogenicity. To achieve this, more effective loading cargo techniques are needed; this is an essential area of research that must be developed further since it would help to establish robust and viable drug delivery approaches and improve and facilitate exosome-based therapies.

**Conclusion and future perspectives.** Exosomes, with their unique cargo signatures, not only play a role in the pathophysiology of many cardiovascular, neurodegenerative and metabolic diseases, but are emerging as useful non-invasive biomarkers for diagnostic and prognostic strategies for several pathological processes. They may also be used as potential drug delivery tools. Exosomes may be potential vectors of therapeutic molecules, and as such may represent an ideal theragnostic approach that would allow us to detect and monitor disease in patients, possibly in the early clinical stages, as well as perform targeted drug delivery at the site of the disease.

While there have been enormous advances in the biology of exosomes, techniques for isolating pure populations of exosomes are still sub-optimal and better approaches to fully characterise exosomes are still required. Nevertheless, the field is growing and there is increasing interest in translating pre-clinical findings to the clinic where exosomes may indeed be the next frontier in novel prognostic, diagnostic and therapeutic tools.

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

## Additional information

### Competing interests

The authors declare that there are no competing interests associated with the manuscript.

### Author contributions

All authors contributed to the writing of this manuscript and approved the final version. All persons designated as authors qualify for authorship, and all those who qualify for authorship are listed.

## Funding

This work was supported by the British Heart Foundation (grant numbers CH/12/429762, RE/13/5/30177, RE18/6/34217 (to R.M.T.)) and the Walton Fellowship, University of Glasgow (to A.C.M.).

## Keywords

biomarkers, cardiovascular system, exosomes, therapeutic target

## Supporting information

Additional supporting information can be found online in the Supporting Information section at the end of the HTML view of the article. Supporting information files available:

**Peer Review History**

