## [Peer Review History · The Journal of Physiology]

Exosomes and the cardiovascular system: role in cardiovascular health and disease

Karla Bianca Neves, Francisco Jose Rios, Javier Sevilla-Montero, Augusto C. Montezano, and Rhian M Touyz
DOI: 10.1113/JP282054

Corresponding author(s): Karla Neves (Karla.Neves@glasgow.ac.uk)

The following individual(s) involved in review of this submission have agreed to reveal their identity: Susan Currie (Referee #1)

Review Timeline:

Submission Date:	30-Nov-2021
Editorial Decision:	14-Jan-2022
Revision Received:	09-Feb-2022
Editorial Decision:	04-Mar-2022
Revision Received:	09-Mar-2022
Accepted:	15-Mar-2022

Senior Editor: Ian Forsythe

Reviewing Editor: Harold Schultz

Transaction Report:

Dear Dr Neves,

Re: JP-TR-2021-282054 "Exosomes and the cardiovascular system" by Karla Bianca Neves, Francisco Jose Rios, Javier Sevilla-Montero, Augusto C. Montezano, and Rhian M Touyz

Thank you for submitting your Topical Review to The Journal of Physiology. It has been assessed by a Reviewing Editor and by 1 expert referee and I am pleased to tell you that it is considered to be acceptable for publication following satisfactory revision.

The reports are copied at the end of this email. Please address all of the points and incorporate all requested revisions, or explain in your Response to Referees why a change has not been made.

NEW POLICY: In order to improve the transparency of its peer review process The Journal of Physiology publishes online as supporting information the peer review history of all articles accepted for publication. Readers will have access to decision letters, including all Editors' comments and referee reports, for each version of the manuscript and any author responses to peer review comments. Referees can decide whether or not they wish to be named on the peer review history document.

I hope you will find the comments helpful and have no difficulty in revising your manuscript within 4 weeks.

Your revised manuscript should be submitted online using the links in Author Tasks Link Not Available. This link is to the Corresponding Author's own account, if this will cause any problems when submitting the revised version please contact us.

You should upload:

- A Word file of the complete text (including any Tables);
- An Abstract Figure, (with accompanying Legend in the article file)
- Each figure as a separate, high quality, file;
- A full Response to Referees;
- A copy of the manuscript with the changes highlighted.
- Author profile. A short biography (no more than 100 words for one author or 150 words in total for two authors) and a portrait photograph of the two leading authors on the paper. These should be uploaded, clearly labelled, with the manuscript submission. Any standard image format for the photograph is acceptable, but the resolution should be at least 300 dpi and preferably more.

- A 'Cover Art' file for consideration as the Issue's cover image;
- Appropriate Supporting Information (Video, audio or data set https://jp.msubmit.net/cgi-bin/main.plex?form_type=display_requirements#supp).

To create your 'Response to Referees' copy all the reports, including any comments from the Senior and Reviewing Editors into a Word, or similar, file and respond to each point in colour or CAPITALS. Upload this when you submit your revision.

I look forward to receiving your revised submission.

Yours sincerely,

Ian D. Forsythe
Deputy Editor-in-Chief
The Journal of Physiology
<https://jp.msubmit.net>
<http://jp.physoc.org>
The Physiological Society
Hodgkin Huxley House
30 Farringdon Lane
London, EC1R 3AW
UK
<http://www.physoc.org>
<http://journals.physoc.org>

EDITOR COMMENTS

Reviewing Editor:

Please see the comments from the referee. A topical review should emphasize state of the art knowledge (new and insightful or controversial) and could benefit with being more forward-looking and present new questions to research. Information that is well accepted and covered in other reviews can be abbreviated to make room for the latest studies on the topic. As noted, figure 1 is not informative and should be improved to better define the distinctions of the various extracellular particles. It was not clear whether there is a distinction between exosomes and extracellular vesicles. The text uses the terms interchangeably. Are there functional differences? Are apoptotic bodies and microparticles worthy of discussion for CV function?

When possible attention should be give to whether exosomes/EVs, as discussed in different scenarios, are serving an autocrine, paracrine or endocrine function. This could be be the focus of a figure illustrating different CV exosome functions.

Senior Editor:

Thank you for this review - it is an important and interesting topic. The referees' feedback is helpful for revision. The graphical abstract is good, but your other figures need a little thought to make sure you provide important information which enlightens your readers about the background, mechanisms and processes involved in this physiology. Please also rewrite your abstract to contain more information about your topic (avoid statements like 'this review highlights....' but instead provide a summary of your findings and come to a discrete final conclusion (avoid saying that more work is required in the abstract, or of making this your conclusion). I look forward to reading your revised manuscript.

REFeree COMMENTS

Referee #1:

This is a timely review of the literature on cardiovascular exosomes and their impact on cell-cell communication as well as their use as potential therapeutics for cardiovascular disease. The biology of exosomes as well as their physiological impact has been well covered but the review would benefit from revision of some of the content and thorough proof-reading.

Revisions that are required are highlighted below.

1. The title should be amended. In the abstract it states that the review focuses on the function of exosomes in CVD but this is not reflected in the title. Perhaps the use of terms Cardiovascular Health and Disease could be considered.
2. In section 1 (the introduction) there is a reference to different patient groups but this is not expanded on - what is meant here? Different gender, race, age or different underlying disease severity?
3. (i) In section 2 there is reference to Figure 1. I feel Figure 1 is pretty vague and much more could be made of this. It's not clear what the text box with 'multiple fluids, non-invasive, real-time monitoring...' is for. Please revise this figure to give more clarity and information.

(ii) Section 2, second paragraph where the molecular mechanisms of biogenesis of exosomes are covered, this would benefit from revision and proof-reading. The three phases should be set out in a consistent manner e.g. (1) early endosome - this starts with.....(2) maturation to late endosome - this is followed by a reduction in..... (3) late endosome - this can either fuse with....
- (iii) Section 2 - there should be a new paragraph from 'The balance of RAB5A and RAB 7A....' and this would benefit from proof reading and an explanatory diagram that could form part of Fig 2.
- (iv) In section 2 the membrane components that participate in exosome interaction with the target cell are mentioned but these are not highlighted in Fig 2 and should be.
4. In section 3 I feel it would make more sense to start with exosome role in health and then move to disease. It may be helpful to sub-section this part to indicate (i) health/cardioprotection (ii) CVD.
5. It is stated that 'some studies have suggested that exosomes have cardioprotective effects but only one reference is quoted (Zamani et al, 2019). Please include additional references.
6. In section 4, the authors refer to Figure 1 to show implication of exosomes in pathophysiological processes but this figure does not reflect that message. Please consider this when revising figure 1.
7. In section 4 the authors should refer to Table 3 in the second paragraph when reviewing exosomes as prognostic markers

of CVD. In the current version, the reference to Table 3 comes later when discussing markers for cancer diagnosis but this does not reflect what is shown in Table 3. This needs to be revised.

8. The authors should also mention cardiotoxicity in section 4. There is reference to a study with doxorubicin but no mention of anti-cancer drug induced cardiotoxicity. It would be pertinent to include this as an area where exosomes may be useful biomarkers.

9. There is mention of 'upregulation' of exosomal miRNA in patients with M.I. but how does this relate to specific biomarkers. Is there upregulation of particular miRNAs? It would be useful to get more context here.

10. I am confused about the content on cancer and neurodegeneration in section 4. This seems out of place in a review devoted to exosomes in cardiovascular health and disease, I can understand reference to stroke and kidney function in the broader context but not cancer or neurodegenerative conditions. Can the authors please revise this content, bearing in mind the title of this section ' Exosomes as biomarkers for cardiovascular injury'?

11. Sections 5 and 6 require careful proof-reading as there are a number of grammatical errors.

REQUIRED ITEMS:

-Please upload separate high quality figure files via the submission form.

-Author profile(s) must be uploaded via the submission form. Authors should submit a short biography (no more than 100 words for one author or 150 words in total for two authors) and a portrait photograph of the two leading authors on the paper. These should be uploaded, clearly labelled, with the manuscript submission. Any standard image format for the photograph is acceptable, but the resolution should be at least 300 dpi and preferably more. A group photograph of all authors is also acceptable, providing the biography for the whole group does not exceed 150 words.

END OF COMMENTS

Confidential Review

30-Nov-2021

RE: Revised Manuscript JP-TR-2021-282054

Manuscript title - Exosomes and the cardiovascular system: role in cardiovascular health and disease

We would like to thank the editor and reviewers for their constructive comments. All concerns have been addressed and we believe the manuscript is significantly improved. Please find herewith the reviewer's comments in black and the answers from the authors in red. We have also highlighted changes in the manuscript in red. We believe that the paper is significantly improved and hope that it is now acceptable for publication.

EDITOR COMMENTS

Reviewing Editor:

Please see the comments from the referee. A topical review should emphasize state of the art knowledge (new and insightful or controversial) and could benefit with being more forward-looking and present new questions to research. Information that is well accepted and covered in other reviews can be abbreviated to make room for the latest studies on the topic. As noted, figure 1 is not informative and should be improved to better define the distinctions of the various extracellular particles. It was not clear whether there is a distinction between exosomes and extracellular vesicles. The text uses the terms interchangeably. Are there functional differences? Are apoptotic bodies and microparticles worthy of discussion for CV function?

When possible, attention should be given to whether exosomes/EVs, as discussed in different scenarios, are serving an autocrine, paracrine or endocrine function. This could be the focus of a figure illustrating different CV exosome functions.

Many thanks for the suggestion. We have modified figure 1 highlighting the role of exosomes as mediators of autocrine, paracrine, and endocrine functions, which now became figure 2 (see page 27). We believe it makes more sense to have the illustration of the exosome biogenesis before bringing the role of exosomes as mediators of intercellular communication. We have also mentioned the role of exosomes as mediators of autocrine, paracrine, and endocrine functions it in the introduction (see page 2). Additionally, as the review is focused on the role of exosomes, we have not included much discussion about apoptotic bodies and microparticles. Most of studies in the literature focus on exosomes and microparticles; not much is discussed about apoptotic bodies.

And finally, as pointed, we have standardized the usage of the word exosomes throughout the text, when appropriate. In few specific cases, authors refer to EVs in general. And this is mainly because the term 'extracellular vesicles' refers to a heterogeneous population of vesicular bodies of cellular origin that derive either from the endosomal compartment (exosomes) or because of shedding from the plasma membrane (microvesicles, and apoptotic bodies), and some authors prefer to use the general term rather than specific subgroups.

Senior Editor:

Thank you for this review - it is an important and interesting topic. The referees' feedback is helpful for revision. The graphical abstract is good, but your other figures need a little thought to make sure you provide important information which enlightens your readers about the background, mechanisms and processes involved in this physiology. Please also rewrite your abstract to contain more information about your topic (avoid statements like 'this review highlights....' but instead

provide a summary of your findings and come to a discrete final conclusion (avoid saying that more work is required in the abstract, or of making this your conclusion). I look forward to reading your revised manuscript.

Many thanks for the suggestion. We have modified figure 1, which now became figure 2 (see page 27). We believe it makes more sense to have the illustration of the exosome biogenesis before bringing the role of exosomes as mediators of intercellular communication. We have also re-written the abstract according to suggestions (see page 2).

REFEREE COMMENTS

Referee #1:

This is a timely review of the literature on cardiovascular exosomes and their impact on cell-cell communication as well as their use as potential therapeutics for cardiovascular disease. The biology of exosomes as well as their physiological impact has been well covered but the review would benefit from revision of some of the content and thorough proof-reading.

Thanks for the constructive comments and suggestions. We have addressed each point carefully.

Revisions that are required are highlighted below.

1. The title should be amended. In the abstract it states that the review focuses on the function of exosomes in CVD but this is not reflected in the title. Perhaps the use of terms Cardiovascular Health and Disease could be considered.

We thank the reviewer for the very constructive suggestion. In consideration, we have changed the title of the review.

2. In section 1 (the introduction) there is a reference to different patient groups but this is not expanded on - what is meant here? Different gender, race, age or different underlying disease severity?

That is a fair comment; we thank the reviewer to point this out. We have clarified the sentence (see page 3). Our aim was to emphasize that, besides the gaps in the understanding of the pathophysiological role of exosomes in CVD, it is also not fully elucidated whether CVD patients with different underlying disease severity present distinct exosome phenotyping.

3. (i) In section 2 there is reference to Figure 1. I feel Figure 1 is pretty vague and much more could be made of this. It's not clear what the text box with 'multiple fluids, non-invasive, real-time monitoring...' is for. Please revise this figure to give more clarity and information.

Many thanks for the suggestion. We have modified figure 1, which now became figure 2 (see page 27). We believe it makes more sense to have the illustration of the exosome biogenesis before bringing the role of exosomes as mediators of intercellular communication.

(ii) Section 2, second paragraph where the molecular mechanisms of biogenesis of exosomes are covered, this would benefit from revision and proof-reading. The three phases should be set out in a consistent manner e.g. (1) early endosome - this starts with.....(2) maturation to late endosome - this is followed by a reduction in..... (3) late endosome - this can either fuse with....

We apologize for the mistake. The corrections were made according to your suggestions (please see page 4).

(iii) Section 2 - there should be a new paragraph from 'The balance of RAB5A and RAB 7A....' and this would benefit from proof reading and an explanatory diagram that could form part of Fig 2.

We have added a new paragraph as suggested (please see page 4). However, we believe adding also the Notch4/ RAB5A/RAB7A/ESCRT signaling would make the figure too overwhelmed as our main aim was to concisely explain the exosomes biogenesis.

(iv) In section 2 the membrane components that participate in exosome interaction with the target cell are mentioned but these are not highlighted in Fig 2 and should be.

In attention to the reviewer's comment, we have revised the figure and have added membrane components that participate in exosome interaction with the target cell and included ESCRT tetraspanins, TSG101 and ALX, which are present in the surface of exosomes and are used as its biomarkers (please see figure 1).

4. In section 3 I feel it would make more sense to start with exosome role in health and then move to disease. It may be helpful to sub-section this part to indicate (i) health/cardioprotection (ii) CVD.

Many thanks for the suggestion. We have changed the text accordingly (please see pages 6-8).

5. It is stated that 'some studies have suggested that exosomes have cardioprotective effects but only one reference is quoted (Zamani et al, 2019). Please include additional references.

We apologize for the mistake. We have included additional references (please see page 6).

6. In section 4, the authors refer to Figure 1 to show implication of exosomes in pathophysiological processes, but this figure does not reflect that message. Please consider this when revising figure 1.

Many thanks for the suggestion. We have changed the reference to table 1 (instead of the figure), where we have the deleterious effects of exosomal miRNA in different CVD, which we believe it is more appropriate (please see page 9).

7. In section 4 the authors should refer to Table 3 in the second paragraph when reviewing exosomes as prognostic markers of CVD. In the current version, the reference to Table 3 comes later when discussing markers for cancer diagnosis but this does not reflect what is shown in Table 3. This needs to be revised.

Thanks for the comment. We have made the changes in the text (please see page 10).

8. The authors should also mention cardiotoxicity in section 4. There is reference to a study with doxorubicin but no mention of anti-cancer drug induced cardiotoxicity. It would be pertinent to include this as an area where exosomes may be useful biomarkers.

In consideration to your important point, we have added a brief discussion about the promising role of exosomes as biomarkers of anti-cancer drugs-induced cardiotoxicity and cardiovascular diseases. (Please see pages 9-10).

9. There is mention of 'upregulation' of exosomal miRNA in patients with M.I. but how does this relate to specific biomarkers. Is there upregulation of particular miRNAs? It would be useful to get more context here.

Thanks for your suggestion. We have added details from the study. Please see pages 9-10 where we have included the following sentence: "A recent study demonstrated that 50 exosomal miRNAs are dysregulated in patients with myocardial infarction. Among them, miR-183 expression was shown to

be significantly upregulated in these patients compared with counterpart control individuals, and its expression was positively associated with the degree of myocardial injury (Zhao *et al.*, 2019).”

10. I am confused about the content on cancer and neurodegeneration in section 4. This seems out of place in a review devoted to exosomes in cardiovascular health and disease, I can understand reference to stroke and kidney function in the broader context but not cancer or neurodegenerative conditions. Can the authors please revise this content, bearing in mind the title of this section 'Exosomes as biomarkers for cardiovascular injury'?

Thanks for your comment. In consideration to it we have removed the paragraphs referring to cancer and neurodegeneration in section 4 (please see page 10).

11. Sections 5 and 6 require careful proof-reading as there are a number of grammatical errors.

Thanks for your comment. We have revised sections 5 and 6 carefully and we hope it reads better.

Dear Dr Neves,

Re: JP-TR-2022-282054R1 "Exosomes and the cardiovascular system: role in cardiovascular health and disease" by Karla Bianca Neves, Francisco Jose Rios, Javier Sevilla-Montero, Augusto C. Montezano, and Rhian M Touyz

Thank you for submitting your Topical Review to The Journal of Physiology. It has been assessed by a Reviewing Editor and I am pleased to tell you that it is considered to be acceptable for publication following satisfactory revision.

The reports are copied at the end of this email. Please address all of the points and incorporate all requested revisions, or explain in your Response to Referees why a change has not been made.

NEW POLICY: In order to improve the transparency of its peer review process The Journal of Physiology publishes online as supporting information the peer review history of all articles accepted for publication. Readers will have access to decision letters, including all Editors' comments and referee reports, for each version of the manuscript and any author responses to peer review comments. Referees can decide whether or not they wish to be named on the peer review history document.

I hope you will find the comments helpful and have no difficulty in revising your manuscript within 4 weeks.

Your revised manuscript should be submitted online using the links in Author Tasks Link Not Available. This link is to the Corresponding Author's own account, if this will cause any problems when submitting the revised version please contact us.

You should upload:

- A Word file of the complete text (including any Tables);
- An Abstract Figure, (with accompanying Legend in the article file)
- Each figure as a separate, high quality, file;
- A full Response to Referees;
- A copy of the manuscript with the changes highlighted.
- Author profile. A short biography (no more than 100 words for one author or 150 words in total for two authors) and a portrait photograph of the two leading authors on the paper. These should be uploaded, clearly labelled, with the manuscript submission. Any standard image format for the photograph is acceptable, but the resolution should be at least 300 dpi and preferably more.

- A 'Cover Art' file for consideration as the Issue's cover image;
- Appropriate Supporting Information (Video, audio or data set https://jp.msubmit.net/cgi-bin/main.plex?form_type=display_requirements#supp).

To create your 'Response to Referees' copy all the reports, including any comments from the Senior and Reviewing Editors into a Word, or similar, file and respond to each point in colour or CAPITALS. Upload this when you submit your revision.

I look forward to receiving your revised submission.

Yours sincerely,

Ian D. Forsythe
Deputy Editor-in-Chief
The Journal of Physiology
<https://jp.msubmit.net>
<http://jp.physoc.org>
The Physiological Society
Hodgkin Huxley House
30 Farringdon Lane
London, EC1R 3AW
UK
<http://www.physoc.org>
<http://journals.physoc.org>

EDITOR COMMENTS

Reviewing Editor:

The authors have done a reasonable job of improving the text, but the abstract and abstract figure need further

improvement. Neither provide a good summary of what the article is about.

In general, I think this is due to lack of focus in the way the authors present the information. These concerns should be easy to correct.

The abstract should relate more to cardiovascular topics in the paper. As written, it does not motivate me to want to read the review

For example: "They play a crucial role in intercellular communication (especially due to their autocrine, paracrine, and endocrine functions) and have been implicated not only [in beneficial effects such as cardiac protection but also] in the pathophysiology of [various] cardiovascular diseases CVD) such as [____,____]". (brackets are suggested changes)

"In addition, exosomes are important non-invasive biomarkers for diagnostic and prognostic strategies for several pathological processes." This is not very informative. Is there an good or important example you discuss to catch reader attention?

"They may also have therapeutic potential, acting as vectors to deliver therapies in a targeted manner. Again this is not very informative of what you actually discuss." It seems there is an emphasis on micro RNA. But it is buried in the text and is not highlighted well, if not for the tables.

Figures: The abstract figure needs to be replaced. It provides no clue what the review is about. It at least needs to center around the cardiovascular system. It reflects the lack of focus in the review. The review focuses on EVs and their physiological roles (benefit), pathophysiological roles (bad), diagnosing CV disease, and curing CV disease. The abstract figure needs to reflect this. As well as making clearer in the abstract and paper.

Similarly, in figure 2, it seems to lack focus to the article. What are the paracrine and endocrine roles played in the CV system? The figure legends seems also ambiguous as to the what is shown. The figure suffers from the same lack of focus discussed above.

Senior Editor:

Thanks for the improvements, but some further efforts are required to better bring your interesting work to the widest audience - as outlined by the RE above.

END OF COMMENTS

1st Confidential Review

09-Feb-2022

RE: Revised Manuscript JP-TR-2021-282054

Manuscript title - Exosomes and the cardiovascular system: role in cardiovascular health and disease

We would like to thank the reviewing editor for the comments. All concerns have been addressed as follows. Please find herewith the reviewer's comments in black and the answers from the authors in red. We have also highlighted changes in the manuscript in red.

Reviewing Editor:

The authors have done a reasonable job of improving the text, but the abstract and abstract figure need further improvement. Neither provide a good summary of what the article is about. In general, I think this is due to lack of focus in the way the authors present the information. These concerns should be easy to correct.

The concerns were addressed as suggested. Please check abstract from revised version.

The abstract should relate more to cardiovascular topics in the paper. As written, it does not motivate me to want to read the review.

For example: "They play a crucial role in intercellular communication (especially due to their autocrine, paracrine, and endocrine functions) and have been implicated not only [in beneficial effects such as cardiac protection but also] in the pathophysiology of [various] cardiovascular diseases CVD) such as [____,____]". (Brackets are suggested changes)

"In addition, exosomes are important non-invasive biomarkers for diagnostic and prognostic strategies for several pathological processes." This is not very informative. Is there an good or important example you discuss to catch reader attention?

"They may also have therapeutic potential, acting as vectors to deliver therapies in a targeted manner. Again this is not very informative of what you actually discuss." It seems there is an emphasis on micro RNA. But it is buried in the text and is not highlighted well, if not for the tables.

Thanks for the suggestions. We have added further information to better define the content of the review.

Figures: The abstract figure needs to be replaced. It provides no clue what the review is about. It at least needs to center around the cardiovascular system. It reflects the lack of focus in the review. The review focuses on EVs and their physiological roles (benefit), pathophysiological roles (bad), diagnosing CV disease, and curing CV disease. The abstract figure needs to reflect this. As well as making clearer in the abstract and paper.

We appreciate the comments. Abstract figure and legend have been changed (see page 25).

Similarly, in figure 2, it seems to lack focus to the article. What are the paracrine and endocrine roles played in the CV system? The figure legends seem also ambiguous as to the what is shown. The figure suffers from the same lack of focus discussed above.

We have changed figure 2 and respective legend according to comments. Note that we kept the different functions (autocrine, paracrine, and endocrine) of exosomes in consideration to previous reviewer's comment (see page 27).

Dear Dr Neves,

Re: JP-TR-2022-282054R2 "Exosomes and the cardiovascular system: role in cardiovascular health and disease" by Karla Bianca Neves, Francisco Jose Rios, Javier Sevilla-Montero, Augusto C. Montezano, and Rhian M Touyz

I am pleased to tell you that your Topical Review article has been accepted for publication in The Journal of Physiology, subject to any modifications to the text that may be required by the Journal Office to conform to House rules.

NEW POLICY: In order to improve the transparency of its peer review process The Journal of Physiology publishes online as supporting information the peer review history of all articles accepted for publication. Readers will have access to decision letters, including all Editors' comments and referee reports, for each version of the manuscript and any author responses to peer review comments. Referees can decide whether or not they wish to be named on the peer review history document.

The last Word version of the paper submitted will be used by the Production Editors to prepare your proof. When this is ready you will receive an email containing a link to Wiley's Online Proofing System. The proof should be checked and corrected as quickly as possible.

All queries at proof stage should be sent to tjp@wiley.com

The accepted version of the manuscript will be published online, prior to copy editing in the Accepted Articles section.

Are you on Twitter? Once your paper is online, why not share your achievement with your followers. Please tag The Journal (@jphysiol) in any tweets and we will share your accepted paper with our 22,000+ followers!

Yours sincerely,

Ian D. Forsythe
Deputy Editor-in-Chief
The Journal of Physiology
<https://jp.msubmit.net>
<http://jp.physoc.org>
The Physiological Society
Hodgkin Huxley House
30 Farringdon Lane
London, EC1R 3AW
UK
<http://www.physoc.org>
<http://journals.physoc.org>

*** IMPORTANT NOTICE ABOUT OPEN ACCESS ***

Information about Open Access policies can be found here <https://physoc.onlinelibrary.wiley.com/hub/access-policies>

To assist authors whose funding agencies mandate public access to published research findings sooner than 12 months after publication The Journal of Physiology allows authors to pay an open access (OA) fee to have their papers made freely available immediately on publication.

You will receive an email from Wiley with details on how to register or log-in to Wiley Authors Services where you will be able to place an OnlineOpen order.

You can check if your funder or institution has a Wiley Open Access Account here <https://authorservices.wiley.com/author-resources/Journal-Authors/licensing-and-open-access/open-access/author-compliance-tool.html>

Your article will be made Open Access upon publication, or as soon as payment is received.

If you wish to put your paper on an OA website such as PMC or UKPMC or your institutional repository within 12 months of publication you must pay the open access fee, which covers the cost of publication.

OnlineOpen articles are deposited in PubMed Central (PMC) and PMC mirror sites. Authors of OnlineOpen articles are permitted to post the final, published PDF of their article on a website, institutional repository, or other free public server, immediately on publication.

Note to NIH-funded authors: The Journal of Physiology is published on PMC 12 months after publication, NIH-funded authors DO NOT NEED to pay to publish and DO NOT NEED to post their accepted papers on PMC.

EDITOR COMMENTS

Reviewing Editor:

The figures and legends have been improved and more informative. There is a typo in the abstract that needs to be corrected. Maybe this can be done in-house?

"Exosomes have been implicated in the pathophysiology of various diseases including cardiovascular disease, cancer (CVD),..."

corrected

"Exosomes have been implicated in the pathophysiology of various diseases including cardiovascular disease (CVD), cancer,..."

[note from Editorial Office: we will make this change for you; please double check at proof stage that all is OK]

Senior Editor:

Thank you for an interesting topical review.